

# Investigating isoform switching in *RHBDF2* and its role in neoplastic growth in breast cancer

Mehar Masood[1,2], Madahiah Bint E Masood[1], Noor Us Subah[1], Maria Shabbir[3], Rehan Zafar Paracha[1] and Mehak Rafiq[1]

[1] School of Interdisciplinary Engineering and Sciences, National University of Sciences and Technology, Islamabad, Pakistan
[2] Faculty of Rehabilitation & Allied Health Sciences, Riphah International University, Islamabad, Pakistan
[3] Atta-ur-Rahman School of Applied Biosciences, National University of Sciences and Technology, Islamabad, Pakistan

Corresponding author
Mehak Rafiq,
mehak@rcms.nust.edu.pk

## ABSTRACT

**Background**. Breast cancer is the second leading cause of cancer-related deaths globally, and its prevalence rates are increasing daily. In the past, studies predicting therapeutic drug targets for cancer therapy focused on the assumption that one gene is responsible for producing one protein. Therefore, there is always an immense need to find promising and novel anti-cancer drug targets. Furthermore, proteases have an integral role in cell proliferation and growth because the proteolysis mechanism is an irreversible process that aids in regulating cellular growth during tumorigenesis. Therefore, an inactive rhomboid protease known as iRhom2 encoded by the gene RHBDF2 can be considered an important target for cancer treatment. Speculatively, previous studies on gene expression analysis of RHBDF2 showed heterogenous behaviour during tumorigenesis. Consistent with this, several studies have reported the antagonistic role of iRhom2 in tumorigenesis, *i.e.*, either they are involved in negative regulation of EGFR ligands via the ERAD pathway or positively regulate EGFR ligands via the EGFR signalling pathway. Additionally, different opinions suggest iRhom2 mediated cleavage of EGFR ligands takes place TACE dependently or TACE independently. However, reconciling these seemingly opposing roles is still unclear and might be attributed to more than one transcript isoform of iRhom2.

**Methods**. To observe the differences at isoform resolution, the current strategy identified isoform switching in RHBDF2 via differential transcript usage using RNA-seq data during breast cancer initiation and progression. Furthermore, interacting partners were found via correlation and enriched to explain their antagonistic role.

**Results**. Isoform switching was observed at DCIS, grade 2 and grade 3, from canonical to the cub isoform. Neither EGFR nor ERAD was found enriched. However, pathways leading to TACE-dependent EGFR signalling pathways were more observant, specifically MAPK signalling pathways, GPCR signalling pathways, and toll-like receptor pathways. Nevertheless, it was noteworthy that during CTCs, the cub isoform switches back to the canonical isoform, and the proteasomal degradation pathway and cytoplasmic ribosomal protein pathways were significantly enriched. Therefore, it could be inferred that cub isoform functions during cancer initiation in EGFR signalling. In contrast, during metastasis, where invasion is the primary task, the isoform switches back to the canonical isoform.

## INTRODUCTION

Breast cancer is the second leading cause of cancer-related deaths globally, and its prevalence rates are increasing day by day *Azamjah, Soltan-Zadeh & Zayeri (2019)*. Unfortunately, cancer treatment is challenging due to the inadequate availability of therapeutic targets (*Mansoori et al., 2017*). In recent years, the introduction of next-generation sequencing and new bioinformatics techniques in genomics and proteomics have made it possible to interact with numerous cancerous genes. This has allowed researchers to see that a gene may evolve into a different polypeptide by being modified at different levels, for example, histone modification and splicing origins, which lead to different isoforms of the same gene having completely altered functioning (*Vitting-Seerup & Sandelin, 2017*). Therefore, there is always an immense need to find promising and novel anti-cancer therapeutic drug targets.

Proteases have an integral role in cell proliferation and growth because the proteolysis mechanism is an irreversible process that aids in regulating cellular growth during tumourigenesis (*Park, Dharmasivam & Richardson, 2020*). Therefore, they can be considered an important target for cancer treatment. Rhomboid proteases are part of the family that hydrolyze the peptide bonds in other proteins and are almost found in all kingdoms of life (*Adrain & Cavadas, 2020*). However, some members of the rhomboid family lack the catalytic residues necessary for proteolysis, suggesting they cannot cleave substrates. Instead, they can do so by complex formation with client proteins known as inactive rhomboids or pseudoprotease (*Bergbold & Lemberg, 2013*). Mammals are reported to have two inactive rhomboids, iRhom1 encoded by the *RHBDF1* gene and iRhom2 encoded by the *RHBDF2* gene. Both share highly conserved protein sequences, and the distinction lies in the protein sequences of the cytosolic region, where they possess different deletions and extensions. Knockout studies on iRhom2 showed more severe phenotypic changes (*Blaydon et al., 2012*; *Hosur et al., 2014*; *Dulloo, Muliyil & Freeman, 2019*). Thus, this makes *RHBDF2* an interesting protein to study. *RHBDF2* has developed a new pseudo enzyme function regulating trafficking, orchestrating inflammatory response and growth factor signalling by interacting with client proteins (*Bergbold & Lemberg, 2013*). Both active and inactive rhomboids have many transcript isoforms in mammals, with several of them that can code for alternative forms of proteins. Whereas iRhom2 have two functionally important isoforms, ENST00000313080 (canonical) and ENST00000591885 (cub), which are also reported in public databases (ENSEMBL and Refseq) along with 18 computationally mapped transcript isoforms.

Several studies have reported the antagonistic role of iRhom2, *i.e.*, either they are involved in negative regulation of EGFR ligands *via* the ERAD (endoplasmic reticulum-associated protein degradation) pathway or positively regulate EGFR ligands *via* the EGFR signalling pathway. Different opinions suggest that the iRhom2 mediated cleavage of EGFR ligands depends on TACE (TNF-α converting enzyme), also known as ADAM17 (*Hosur et al.,*
*2014*; *Künzel et al., 2018*). In contrast, some research suggests that an independent pathway of TNF-α exists. Therefore, iRhom2 may only serve as a catalyst for TNF secretion, or it may be functionally redundant with other proteins (s) (*Siggs et al., 2014*). Furthermore, evidence has shown that the onset of sleep-like phenotype in *Drosophila melanogaster* is due to iRhoms being involved in the negative regulation of EGFR signalling through the ERAD pathway in the nervous system (*Lee, Nam & Choi, 2016*).In contrast, active rhomboids regulate the cleavage of EGFR membrane-bound precursors (*Adrain & Freeman, 2012*). Some conserved mechanistic links exist between mammals and drosophila in the regulation of EGFR signalling and in maintaining cell quality control machinery for efficient trafficking (*Etheridge et al., 2013*). iRhom2 can negatively regulate EGFR signalling *via* the breakdown of EGF-like substrates. They increase ERAD activity by bringing clients passively by delaying endoplasmic reticulum (ER) retention, enhancing the chance of exposure to ERAD machinery (*Lee, Nam & Choi, 2016*). While the Freeman Research Group in 2011 suggested that they can perform this mechanism by specifically destabilizing some substrates in ER, inhibiting their access to active rhomboids and leading to degradation (*Zettl et al., 2011*). Apart from cancer, high expression of iRhom2 in renal tubules has been identified as the target of PPAR $\gamma$, thus promoting EGF degradation *via* ERAD (*Lyu et al., 2018*). Further work was done on TACE-independent mediated regulation of EGFR ligand (*Hosur et al., 2018*). The study performed conditional deletion of ADAM17, in RHBDF2 impaired amphiregulin (AREG) mediated sebaceous gland enlargement, wound healing and alopecia suggesting ADAM17 is essential for shedding of EGFR ligand.

Studies on breast cancer have stated that iRhoms can regulate proliferation during tumorigenesis *via* GPCR (G-protein coupled receptor) signalling by transactivation of EGFR signalling (*Christova et al., 2013*). These pseudoproteases are essential for the maturation and trafficking of ADAM17 to the plasma membrane from ER through the Golgi apparatus and are also linked to the fates of TNF-α and EGFR ligands (*Lee, Nam & Choi, 2016*). Consistent with this, *in silico* analysis of publicly available gene expression data-sets on breast cancer showed heterogeneous expression behaviour of *RHBDF2* according to the intrinsic molecular subtypes and histopathological grading and staging (*Canzoneri et al., 2014*). However, progression to carcinoma is not as simple, and we found no literature for studying mRNA expression of *RHBDF2* during neoplastic growth, *i.e.*, cancer initiation and progression. ICD-10 classifies neoplasms into four groups, benign neoplasm, *in situ* neoplasm, malignant neoplasm and neoplasm of uncertain or unknown behaviour (*WHO, 2016*). Studies state that an alternative splicing mechanism is pathologically altered during neoplastic growth, impacting the cell behaviour and causing tissue-specific changes (*Lu et al., 2015*; *Chabot & Shkreta, 2016*). Alternative splicing and its related proteins are anticipated to be involved in the dynamic phenotypic changes in cancer cells (*Chabot & Shkreta, 2016*). The importance of analyzing isoforms instead of genes has been highlighted because cancer cell growth is directly linked to the aberrant use of one alternatively spliced formed isoform over another under unfavourable circumstances (*Soneson, Love & Robinson, 2020*). Hence, the isoform switching might explain the heterogenous expression of iRhom2, which leads to its divergent roles as stated above in either the ERAD or EGFR pathway.

## Proposed work

In the present study, we explore the antagonistic role and heterogenous expression behaviour of *RHBDF2* encoding iRhom2 during the neoplastic growth in the breast, considering the bidirectional role that might be attributed to the presence of more than one functionally important isoforms of *RHBDF2* in mammals during tumorigenesis.

## METHODS

### Data Collection

The paired-end fastq files containing raw reads for samples in each dataset (GSE52194, GSE130660, GSE69240, GSE110114, GSE45419, GSE51124, GSE148991) were downloaded using an FTP link from EMBL-EBI (The European Bioinformatics Institute) and ENA (European Nucleotide Archive), having accession numbers followed by the SRR acronym. The datasets were classified as follows (GSE52194, GSE130660 for normal *versus* primary tumour), (GSE69240 for normal *versus* ductal carcinoma *in-situ* DCIS), (GSE110114, GSE45419 for normal *versus* invasive ductal carcinoma- IDC), (GSE148991 for normal *versus* circulating tumour cells-CTCs) and (GSE51124 for normal *versus* grade2 and grade3) also shown in Table S1.

### Transcriptome reconstruction and quantification

The raw data were pre-processed for adapter sequences using fastp (*Soneson, Love & Robinson, 2020*). The transcriptome data was then analyzed using the new tuxedo pipeline (*Pertea et al., 2016*). First, the filtered reads were aligned on the reference genome GRch38 using Hisat2. Next, the mapped reads from each sample and the genome GTF file were used to perform annotation-based transcriptome assembly using StringTie. The assemblies were then compared and merged. The StringTie merge function creates a set of merged transcripts comparable to the subsequent analysis.

### Differential isoform expression

StringTie produces a set of reads or coverage tables/files of the quantified or abundance data that were read into R version 3.6.2 for isoform expression analysis using Bioconductor package IsoformSwitchAnalyzeR version 1.8.0. Importing data includes preparing a transcript sequence FASTA file, a parent directory containing coverage table/files, quantification files of the samples in GTF format, and a design file enlisting the phenotypic data, *i.e.*, sample ID and its corresponding condition. The FASTA sequence file for transcripts was generated using a program utility called gffread (*Pertea & Pertea, 2020*). The utility generates a FASTA file with DNA sequences for all the transcripts in the GTF file. The inter-sample normalisation was done using edgeR embedded in the importRdata() function that concatenates all the information into SwitchAnalyzeRlist. This SwitchAnalyzeRlist is an object containing all the data frames and phenotypic data related to the dataset. The abundance files generated *via* StringTie are already normalised for intra-sampling using the FPKM approach. The EdgeR works best for inter-sample normalisation *via* the TMM (trimmed mean of M values) method on pre-normalised count data (*Maza, 2016*). The normalised data were prefiltered to remove the uninterested data of transcripts and genes

from the switch list object, such as non-expressed isoforms or genes and genes with only one isoform. The differential isoform usage test was performed using the function isoformswitchTestDEXSeq(), enabling the switch identification.

## Annotating unknown transcript isoform

During the stringTie-merge step, transcripts are labelled as MSTRGs. These could sometimes be either novel transcripts, false positives or valid transcripts that are left unannotated. The transcripts with MSTRGs labels were annotated *via* BLASTp. To ensure whether they are novel or already exist and are mislabeled. BLASTp checks the sequence similarity of these MSTRGs to already annotated transcript sequences deposited in the public databases. The input sequence of these MSTRGs for BLASTp was extracted *via* the extract sequence function(). The MSTRGs were further considered for downstream analysis based on $E$-value cut-off $= 0$, per cent identity; *i.e.*, 100% and query length matching the length of the already existing annotated transcript, *i.e.*, the target sequence length. For canonical transcript (ENST00000313080), it should be 856 aa; for cub transcript (ENST000591885), it should be 827 aa. The final transcripts were analysed to predict their functional consequences using external tools. CPC2 (*Jian Kang et al., 2017*) was used to check the coding potential, and Pfam (*Punta et al., 2012*; *Finn et al., 2016*) to predict the biological domains. The input FASTA files for the tools were manipulated using the Seqkit (*Shen et al., 2016*) package according to the requirements of each tool.

## Finding interacting partners *via* correlation

The interacting partners were found using Pearson correlation. prepDE.py python script was used to obtain read count information from the quantification file generated *via* StringTie.The count files were then subjected to differential analysis using DESeq2 (*Love, Huber & Anders, 2014*). The normalised differential counts were then input for correlation analysis. Pearson correlation was used to find the most correlated genes with RHBDF2 at the significance level level $p$-value $<0.05$ and correlation cutoff $\pm 0.7$.

## Enrichment

The most significant correlated partners were enriched using GSEA (gene set enrichment analysis) following the protocol outlined earlier in *Subramanian et al. (2005)*.

# RESULTS

## Annotation of MSTRGs *via* BLASTp

To ensure whether MSTRGs labelled transcripts are novel or already exist and are mislabeled, BLASTp was used to annotate them *Acland et al. (2013)*. If any of these transcripts matched the already-annotated transcripts based on the query length, per cent identity, and $E$-value, they would be merged with already-annotated transcripts for further analysis. The MSTRG labelled transcripts (MSTRG.15617.1, MSTRG.15617.2, MSTRG.15617.3) from the dataset GSE110114 of query length 827 amino acid (AA) showed 100% identity with isoform-2 of *RHBDF2* (accession id NP_001005498.2), *i.e.*, length 827 AA exactly matching the query length, and $E$-value 0. Similarly, following

MSTRG labelled transcripts fulfilled the above mentioned criteria: the transcripts (MSTRG.20860.1, MSTRG.20860.2, MSTRG.20860.3, MSTRG.20860.4) from the dataset GSE69240, the transcripts (MSTRG.17267.4 and MSTRG.17267.5) from the dataset GSE52194, the transcripts (MSTRG.17431.2, MSTRG.17431.3, MSTRG.17431.4, MSTRG.17431.5, MSTRG.17431.8, MSTRG.17431.9, MSTRG.17431.10) from the dataset GSE45419, the transcript (MSTRG.15555.1) from the dataset GSE130660, the transcripts (MSTRG.19489.1,MSTRG.19489.3, MSTRG.19489.4, MSTRG.19489.5, MSTRG.19489.8) from the dataset GSE51124, the transcripts (MSTRG.22199.1, MSTRG.22199.5, MSTRG.22199.7, MSTRG.22199.9) from the dataset GSE148991. These transcripts were further selected for analysis also shown in Table 1.

Similarly, the following showed 100% identity with isoform-1 of *RHBDF2* (accession id NP_078875.4). From the dataset GSE52194, the transcript (MSTRG.17267.1), from the dataset GSE45419, the transcripts (MSTRG.17431.1, MSTRG.17431.13), from the dataset GSE148991, the transcripts (MSTRG.22199.2, MSTRG.22199.10) These transcripts were further selected for analysis also shown in Table 2.

## Cross-validation of BLASTp hit sequences

The MSTRGs transcripts showed 100% identity and similarity match hit with NCBI sequences using BLASTp. The NCBI protein sequence database sources are RefSeq and Genbank mainly. ENSEMBL gene sets are derived from multiple sources, partly from RefSeq, partly from uniport and partly from Havana annotation. Although, as stated above, the criteria have been set for matching query length using ENSEMBL annotated transcripts, it was essential to find whether a transcript from ENSEMBL closely matches the transcript from BLASTp hit. To cross-validate the ENSEMBL annotated sequences to the corresponding BLASTp annotated sequences, multiple sequence alignment was performed *via* CLUSTAL. The FASTA sequences were extracted from the NCBI for accession id (NP_001005498.2 and NP_078875.4) corresponding to selected MSTRG transcripts. The alignment showed 100% similarity between NP 001005498.2 and ENST0000591885 (cub) and 100% similarity between NP 078875.4 and ENST0000313080 (canonical) shown in File S1. Thus, selected MSTRGs were merged with annotated transcripts and used further for calculations.

## Isoform expression

Figure 1 shows the isoform expression *via* parameter isoform fraction of the two transcript isoforms canonical and the cub across the two conditions, normal shown by green bars and tumour shown by orange bars. Isoform fraction is calculated using the ratio between isoform expression and gene expression values in each condition. The isoform expression in normal *versus* primary tumour conditions (GSE52194 and GSE130660) of both the transcripts (canonical and cub) significantly decreases. Whereas, for the dataset (GSE110114 and GSE45419) where the comparison is between normal *versus* IDC conditions, the change in the isoform expression is insignificant. Interestingly for the dataset (GSE69240) normal *versus* DCIS, (GSE51124) normal *versus* grade 2 and grade 3, and (GSE148991) normal *versus* CTCs, the change in isoform expression is statistically significant.

**Table 1  MSTRG labelled transcripts showing similarity match with cub isoform.**

| Dataset | Similarity match with cub isoform |
|---|---|
| GSE52194 | MSTRG.17267.4, MSTRG.17267.5 |
| GSE130660 | MSTRG.15555.1 |
| GSE69240 | MSTRG.20860.1, MSTRG.20860.2, MSTRG.20860.3, MSTRG.20860.4 |
| GSE110114 | MSTRG.15617.1,MSTRG.15617.2,MSTRG.15617.3 |
| GSE45419 | MSTRG.17431.2, MSTRG.17431.3, MSTRG.17431.4, MSTRG.17431.5, MSTRG.17431.8 MSTRG.17431.9, MSTRG.17431.10 |
| GSE51124 | MSTRG.19489.1, MSTRG.19489.3, MSTRG.19489.4, MSTRG.19489.5,MSTRG.19489.8 |
| GSE148991 | MSTRG.22199.1, MSTRG.22199.5, MSTRG.22199.7, MSTRG.22199.9 |

**Table 2  MSTRG labelled transcripts showing similarity match with canonical isoform.**

| Dataset | Similarity match with canonical isoform |
|---|---|
| GSE52194 | MSTRG.17267.1 |
| GSE45419 | MSTRG.17431.1, MSTRG.17431.13 |
| GSE148991 | MSTRG.22199.2, MSTRG.22199.10 |

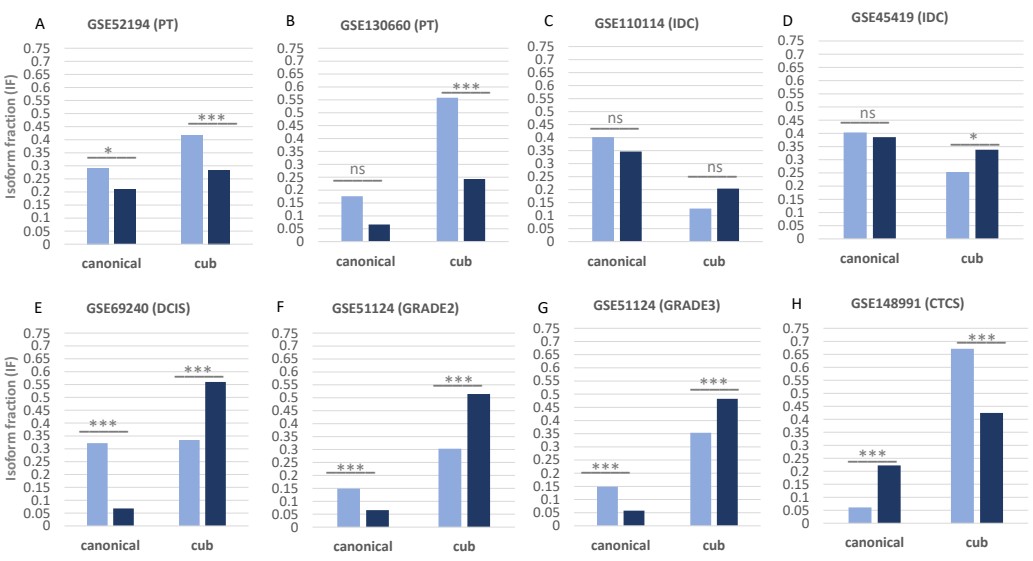

**Figure 1  Isoform fraction of canonical and cub transcript in normal and tumor condition.** (A–H), Normal condition is shown *via* green bars and tumor condition *via* orange bars. The normal *versus* primary tumor dataset shows that isoform fraction of both the isoforms canonical and cub are significantly less in tumor state whereas in normal *versus* IDC the change in isoform fraction is insignificant. For normal *versus* DCIS, grade2 and grade3 isoform fraction of cub isoform in tumor state is significantly more than canonical isoform. Interestingly, in normal *versus* CTCs cub isoform fraction start decreasing significantly.

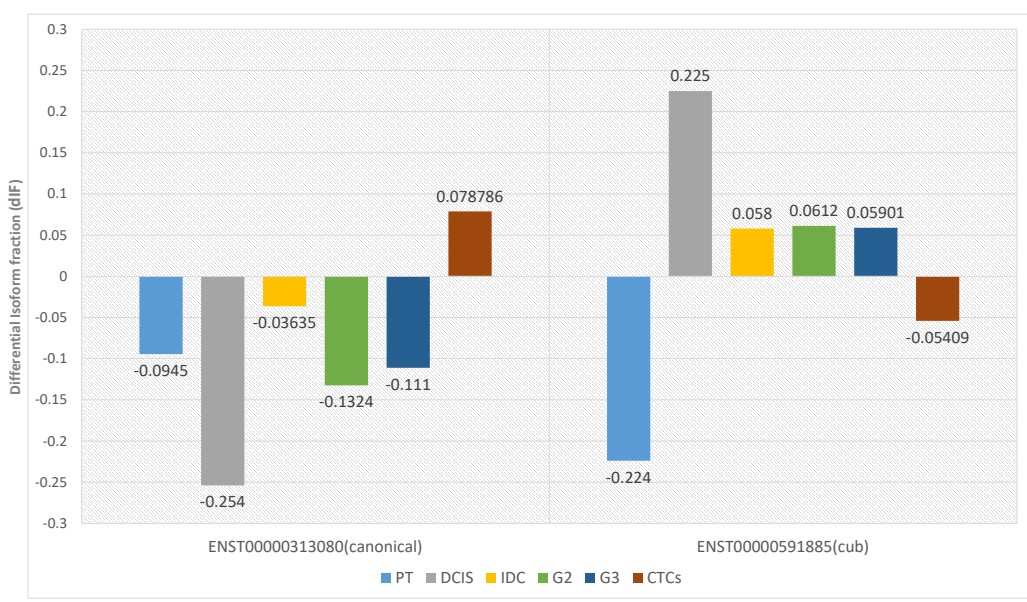

**Figure 2 Differential isoform fraction of canonical and cub transcript.** The blue bar represent that both the isoforms canonical and cub showed decreased usage. During DCIS (gray), grade2 (green), grade3 (dark blue) canonical isoform shows decreased usage whereas cub isoform shows increased usage. During IDC (yellow) although there is an opposite usage but unable to fulfill dIF cutoff. Interestingly, during CTCs (brown bar) isoform expression is reverted where cub isoform shows decreased usage and canonical isoform shows increased usage.

## Differential isoform usage

Two parameters are considered for significant isoform switching, *i.e.*, the statistical significance and the effect size. Statistical significance is calculated *via* $p$-value, and it should be $<0.05$. Whereas effect size tells the association between the two variables, differential isoform fraction (dIF) measures the change in the isoform fractions of the pair of isoforms across the condition. For isoform switching, the pair of isoforms in each dataset should show an opposite increase or decrease in the isoform usage across the conditions. It is calculated by taking the difference between the isoform fraction values. The cut-off for dIF $\geq 0.05$. Figure 2 shows that the isoform switches from canonical to the cub transcript isoform at DCIS, grade2 and grade3, whereas it switches back to the canonical isoform at CTCs.

## Relative isoform fraction

The analyses were done on two primary tumour datasets and two IDC datasets, hence to make a better inference, the isoform fraction (IF) values of the two primary tumour datasets and two IDC datasets were merged by taking the average values. However, a meta-analysis was not done as it has been seen in the literature that the merging approach also gives comparable results after the list of DEGs has been obtained, in our case DTUs (*Taminau et al., 2014*). Therefore, the final plot in Fig. 3 is constructed, showing the relative usage of the isoform fractions of the two isoforms across conditions. Here, in normal *vs* primary tumour, the relative change of the two isoforms across two conditions remains the same,

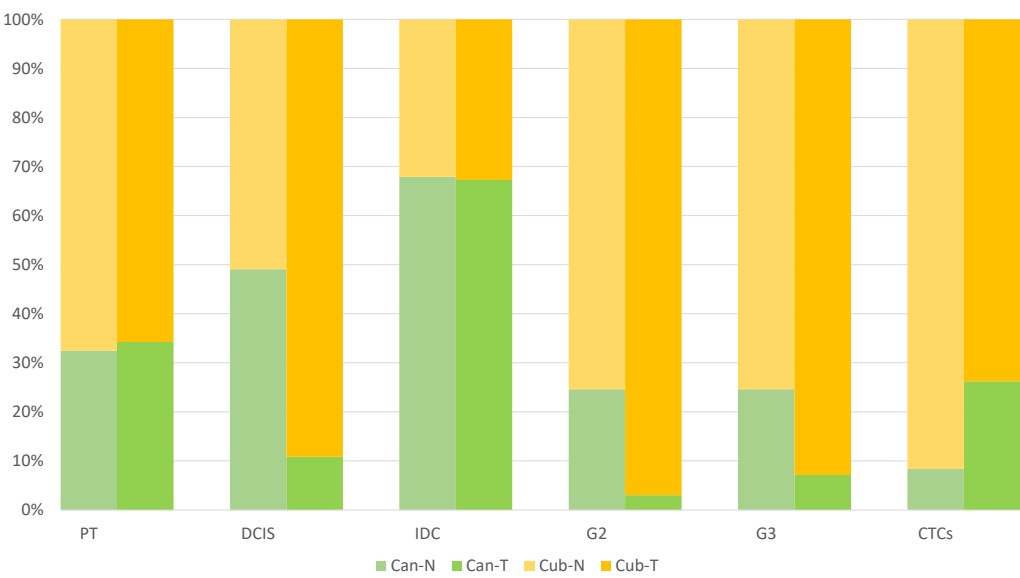

**Figure 3** **The plot shows the relative isoform usage of the canonical and cub isoform in normal and tumor conditions.** The light green bar represents canonical isoform usage in normal conditions (Can-N), and the dark green bar represents canonical isoform usage in tumor conditions (Can-T). In contrast, the light orange bar represents cub isoform usage in normal condition (Cub-N) and the dark orange bar represents cub isoform usage in the tumor condition (Cub-T).

*i.e.*, in normal conditions, canonical is 30%, and the cub is 70% similarly, in tumour conditions, canonical is 30%, and the cub is 70%. However, in normal *vs* DCIS, it can be seen that in normal conditions, the contribution of the canonical and cub transcript to the overall gene expression is equal, *i.e.*, 50%. In contrast, in tumour conditions, cub transcript expression rises to 90% of the overall gene expression relative to the canonical transcript, which is just 10%. Hence, it can be said that a switch in expression has occurred.

Surprisingly, the same trend can be seen in normal *vs* IDC, like normal *vs* primary tumour. The relative change of the two isoforms across two conditions is the same, *i.e.*, in normal conditions, canonical is 68%, and the cub is 32%. However, in normal *vs* grade 2, in normal conditions, the relative usage of the canonical transcript is 22%, and the cub transcript is 78%.In contrast, there is a drastic change in relative usage across the conditions. Here, the cub transcript contributes 94% to the overall gene expression compared to canonical, which is 6%. Similarly, in normal *vs* grade 3, in normal conditions, the relative usage of the canonical transcript is 22%, and the cub transcript is 78%. In comparison, in tumour conditions, the relative usage of the canonical transcript is 8%, and the cub transcript is 92%. Finally, in normal *vs* CTCs, in normal conditions, the relative usage of the canonical transcript is 9%, and the cub transcript is 91%. In comparison, in tumour conditions, the relative usage of the canonical transcript is 28%, and the cub transcript is 72%.

Hence, it is clear from the plot that isoform switching from canonical to cub transcript exists at DCIS, grade 2 and grade 3, while at CTCs, the transcript switches from cub to canonical.

## Finding interactive partners *via* correlation

Isoform switching was observed at DCIS, grade 2, grade 3 and CTCs, so it was essential to find the interacting gene partners *via* correlation analysis. Correlation analysis was performed at +0.7 to −0.7 cut-off to find the statistically significant correlated gene partners. Interacting genes were then subjected to enrichment analysis *via* GSEA (*Subramanian et al., 2005*). GSEA determines a defined set of genes and their biologically meaningful interpretation across two conditions or phenotypes. Since our interacting genes were already ranked according to the correlation criteria, the analysis was done using pre-ranked GSEA. Pre-ranked GSEA was run on default parameters except for the minimum gene set size parameter set to 5, *i.e.*, gene sets smaller than five were excluded from the analysis. First, the union of interacting gene partners and their correlation values were taken as input for pre-ranked GSEA to improve the diverse genomic information; secondly, canonical switched to the cub in all conditions. Since, during CTCs, the cub isoform switches back to the canonical isoform in tumour conditions, a list of the ranked gene for CTCs was made separately.

The biological and molecular processes corresponding to mitogen-activated protein kinase (MAPK), G-protein coupled receptor (GPCR) and toll-like receptor-related signalling pathways were most commonly observed among all the databases. Table 3 shows the GSEA results for the interacting partners for DCIS, G2 and G3 (Files S2 and S3) and Table 4 shows for CTCs along with *p*-value and the number of the genes in an enriched gene-set (Files S4 and S5).

## Running leading edge analysis

Not all the members in a gene-set are particularly contributing to the biological pathway. Therefore, extracting the core genes that contribute more to the enrichment score of the significant biological pathways is often useful. The leading-edge subset in a gene set is those genes that appear in the ranked list at or before the point at which enrichment score (ES) reaches its maximum deviation from zero. After running GSEA, leading-edge analysis helps to examine the genes in the leading-edge subsets of the enriched gene sets. A gene in many leading-edge subsets is more likely to be of interest than a gene in only a few leading-edge subsets. The subset of genes from the leading-edge analysis is shown in Fig. 4.

The heatmap shows the names of those gene subsets found mainly enriched in one or all gene sets. For example, *RELA, RELB, RRAS, GNA12,* and *PRKACA* are the interesting genes found in 2 out of 3 gene sets for DCIS, grade 2 and grade 3, whereas *NFKB1* is the only exciting gene found in two out of five gene sets for CTCs. Here, gene sets correspond to the biological process that is enriched.

## DISCUSSION

Breast cancer is the leading cause of oncologic mortality and morbidity among women worldwide. Mostly all breast carcinomas appear to originate from the uncontrolled

**Table 3 Enriched biological processes for the most correlated genes with RHBDF2 during DCIS, grade 2 and grade 3.**

| Biological processes | Size | *p*-value |
|---|---|---|
| MAPK signaling pathway | 38 | 0 |
| G protein signaling pathways | 12 | 0.008475 |
| Toll-like receptor signaling pathway | 6 | 0.00352 |

**Table 4 Enriched biological processes for the most correlated genes with RHBDF2 during CTCs.**

| Biological processes | Size | *p*-value |
|---|---|---|
| MAPK signaling pathway | 16 | 0.047 |
| G protein signaling pathways | 9 | 0.043 |
| Toll-like receptor signaling pathway | 5 | 0.006 |
| Proteasome degradation | 7 | 0.0026 |
| Cytoplasmic ribosomal proteins | 17 | 0.004 |

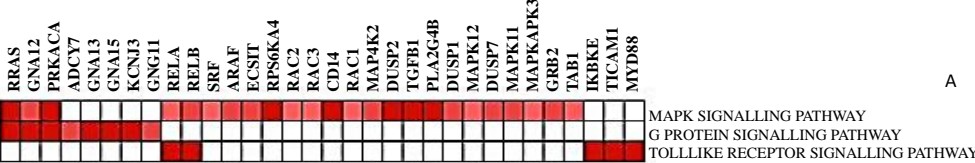

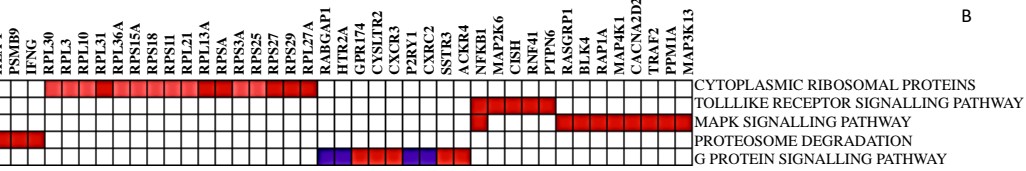

**Figure 4 The heat map shows the (clustered) genes in the leading-edge subsets.** Heatmap (A) is for DCIS, grade 2, and grade 3, whereas heatmap (B) is for CTCs. The range of colors (red, pink, light blue, dark blue) shows the range of correlation values (high, moderate, low, lowest) in an enriched gene-set.

production of epithelial cells of breast tissues forming a lump, as shown in Fig. 5. When normal epithelium begins to undergo malignant transition, the first progressive phase of excessive proliferation known as hyperplasia occurs, followed by the appearance of aberrant cells. At a later phase, known as carcinoma *in situ*, these cells acquire a malignant phenotype but lack invasive properties due to the loss of cell motility. These cancer cells then grow into a solid tumour, eventually causing new blood vessels to grow, undergoing angiogenesis. In the final phase of progression, the cell undergoes complete morphological changes causing the cells to break through basal membranes, thereby becoming invasive carcinoma (*Allred, Mohsin & Fuqua, 2001*). Next, they invade through EMT (epithelial to mesenchymal transition), a process called invasion, and enter the bloodstream (intravasation), leading

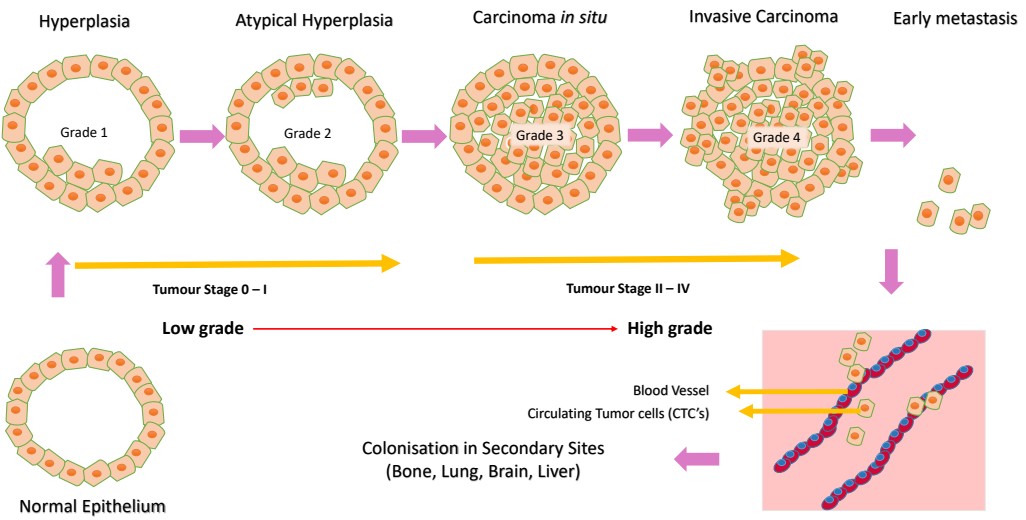

**Figure 5  Cancer initiation and progression.** Steps in neoplastic growth during breast cancer.

to tumour metastasis. These circulating tumour cells are called CTCs (circulating tumour cells) (*Harbeck et al., 2019*). This process is illustrated in Fig. 5 addition, studies have shown that splicing is often pathologically altered, impacting cell behaviour during cancer initiation and progression (*Lu et al., 2015*; *Chabot & Shkreta, 2016*).

More than 90% of the eukaryotic genes in mammals generate multiple isoforms, and aberrant splicing has become the cause of many human diseases (*Sorek, Shamir & Ast, 2004*). Typically, coding genes have a transcript isoform expressed significantly higher than other alternatively spliced transcript isoforms, often known as canonical isoforms. However, under unfavourable circumstances like disease states, the dominance may completely shift from canonical to the other alternative transcript isoforms (*Di et al., 2018*).

Several studies have reported the antagonistic role of iRhom2 in tumorigenesis and other diseases, *i.e.*, either they are involved in negative regulation of EGFR ligands *via* the ERAD pathway or positively regulate EGFR ligands leading to the EGFR signalling pathway. Furthermore, parallel studies suggest iRhom mediated cleavage of EGFR ligands *via* TACE-dependent or TACE-independent pathway (*Al-Salihi & Lang, 2020*). Therefore, it can be hypothesised that the controversial role may be attributed to more than one active isoform of iRhom2 in performing alternative physiological activities in the cell. Our study tested this hypothesis, and we report that isoform switches from canonical to the cub transcript isoform at DCIS, grade 2, grade 3 and from cub to canonical transcript isoform at CTCs during neoplastic growth. Gene enrichment and Pearson correlation showed that during the isoform switching in DCIS, grade 2, grade 3, the biological processes leading to TACE-dependent EGFR pathway were enriched, *i.e.*, MAPK signalling pathway, GPCR pathway and toll-like receptor pathways. Leading-edge analysis showed *RELA, RELB, RRAS, GNA12* and *PRKACA* were the gene found among 2/3 of the biological processes for DCIS, grade 2 and grade 3. In contrast, *NFKB1* was found in 2/5 of the biological processes for CTCs.
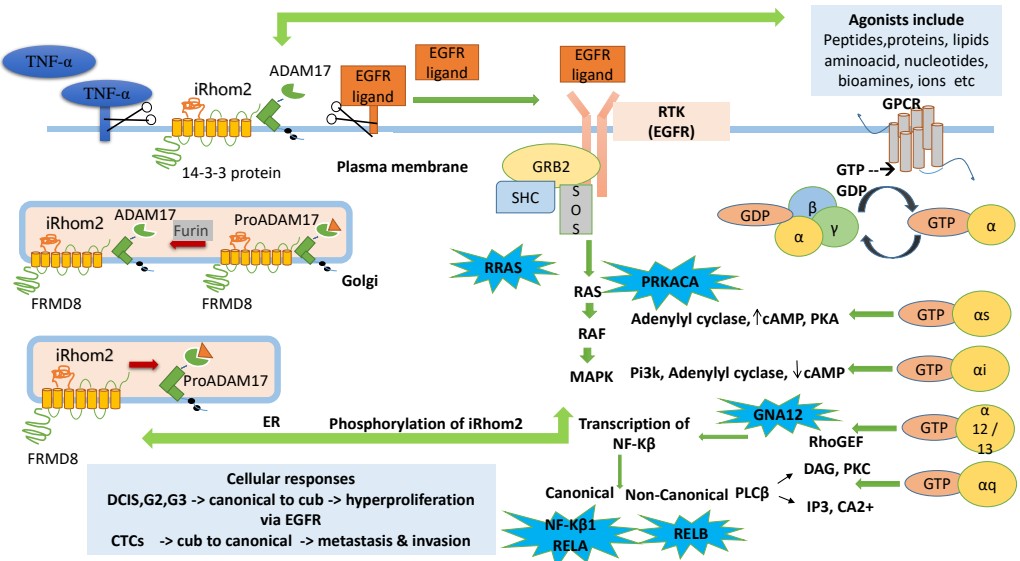

**Figure 6  Cub pathway.** The figure shows the proposed mechanistic links between signaling pathways GPCR, MAPK, RTK (EGFR) and the role of *ADAM17* (TACE) dependent RHBDF2 (iRhom2) during tumorigenesis. The enriched genes (*PRKACA, RRAS, GNA12, NF-κ B1, RELA, RELB*) in the leading-edge analysis are highlighted in blue bubbles and shown at the site involved during downstream signaling.

A tentative pathway is proposed to explain the role of iRhom2 from the analysis carried out in Fig. 6. Previously it is known that TACE is synthesised in ER as an immature form containing an inhibitory prodomain that prevents its proteolytic activity. iRhom2 forms a complex with FRMD8 protein, which is its interacting protein. This complex, along with enzyme furin, helps remove the prodomain and converts the TACE into an active form in Golgi. The mature TACE traffics to the plasma membrane with iRhom2. The binding of 14-3-3 proteins to the iRhom2 N-terminal domain weakens the interaction with TACE at the cell surface. The TACE at the plasma membrane cleaves the EGFR and TNF-α ligands *via* its sheddase activity, thereby mitigating the onset of signalling and inflammatory pathways (*Dulloo, Muliyil & Freeman, 2019*). Without iRhoms, there is no TACE maturation and no TACE activity. Mounting literature and evidence from physiological and molecular data show that the alterations in TACE function are due to evident mutations or deletions in N-terminal (*Blaydon et al., 2012*; *Brooke et al., 2014*; *Siggs et al., 2014*; *Li et al., 2015*).

GPCR agonists activate ADAM metalloproteases (ADAM10, ADAM12, ADAM17) to produce mature EGFR ligands leading to EFGR transactivation. It has been well documented that EGFR transactivation *via* GPCR plays a crucial role in proliferation and migration-associated physiological functions (*Prenzel et al., 1999*; *Wang, 2016*). The GPCR signalling pathway begins with activating the receptors with suitable agonists (ligand activation) and ends with the downstream regulation of various cellular processes such as proliferation, migration, angiogenesis, differentiation, and survival. Activated *via* agonists (lipids, proteins, amino acids, bio-amines, nucleotides, hormones, or neurotransmitters),

GPCRs function by interacting with intracellular G-proteins. They are formed by combining three subunits, $\alpha, \beta$ and $\gamma$. They are identified *via* G $\alpha$ monomers and further grouped into four families (G $\alpha$s, G $\alpha$i, G $\alpha$q and G $\alpha$12) (*Smrcka, 2008*; *Senarath et al., 2018*; *Weis & Kobilka, 2018*; *Wu & Gutkind, 2020*).

*PRKACA* identified in this study helps in the phosphorylation of different enzymes and proteins. cAMP-dependent phosphorylation of proteins is essential to many cellular processes, including differentiation, proliferation, and apoptosis. Several mutations in this gene promote various cancers (*Moody et al., 2014*). GNA12 is an $\alpha$ guanine nucleotide-binding protein, and these heterotrimeric subunits link GPCRs to the nucleotide exchange factors, which interact with Rho GTPases that regulate cell invasion in breast cancer (*Chia, Kumari & Casey, 2014*). The research also states that the activation of *GNA12* in BC stimulates the promotor activity *via* NF-κB binding of interleukins and matrix metalloproteinase (MMP-2).

*RELA*, another correlated gene with iRhom2, is a protein-coding gene known as the p65 transcription factor and NF-κB subunit. NF-κB is a transcription factor involved in several biological processes like cell growth, inflammation, tumorigenesis, immunity and apoptosis. It is ubiquitous, *i.e.*, present in an inactive form in the cytoplasm by specific inhibitors; upon degradation of these inhibitors, NF-κB moves to the nucleus and regulates specific genes. NF-κB comprises NF-κB1 or NF- κB2 bound to either subunit REL, RELA or RELB *Chaturvedi et al. (2011)*. RELA-NF-κB1 appears to be the most abundant complex. RELA is expressed in many cells like epithelial, neuronal, endothelial, and activation of this gene is positively correlated with multiple cancers. Post-transcriptional modification like methylation is associated with NF-κB1 in Breast cancer (*Jeong, Oh & Choi, 2019*). RELB is found to be expressed at higher levels in Breast cancer in regulating the noncanonical NF-κB pathway. It promotes cell proliferation and enhances cell motility by activating EMT (*Wang et al., 2020*). These signaling pathways are initiated by the binding of extracellular growth factors (ligands/ signaling molecules) to transmembrane receptor tyrosine kinases (RTKs) such as EGFR. RTKs are linked indirectly to Ras *via* two proteins, GRB2 and Sos. Ras cycles between an inactive GDP-bound form and active GTP-bound form. Ras cycling requires the assistance of two proteins, GEF and GAP. The SH2 domain in GRB2, an adapter protein, binds to specific phosphor-tyrosines in activated RTKs. The two SH3 domains in GRB2 then bind Sos, a guanine nucleotide exchange factor, thereby bringing Sos close to membrane-bound Ras- GDP and activating its exchange function. Binding of Sos to inactive Ras causes a large conformational change that permits release of GDP and binding of GTP. RAS becomes active converting GDP to GTP leading to the activation of RAF and MAPK signalling pathways (*Cussac, Frech & Chardin, 1994*).

*RRAS* correlated to the cub isoform of *RHBDF2* is a small GTPase binding protein. It is involved in angiogenesis, cell adhesion, neuronal regulation and vasculogenesis. Recently, a negative association exists between activation of the *RRAS* gene and breast cancer progression, and loss of activation of this gene leads to carcinogenesis (*Song et al., 2014*). The Ras then leads to the activation of RAF and MAPK signaling pathway. MAPK signaling pathway then phosphorylates the iRhom2 N-terminal domain. iRhom2 binding with ADAM17 controls several aspects of its activity, including stimulated shedding

activity on the cell surface. ADAM17 shedding stimuli triggers MAP kinase-dependent phosphorylation of iRhom2 N terminal cytoplasmic tail. The regulation of sheddase activity at the cell surface is controlled *via* several stimulatory agents like G protein-coupled receptors, toll-like receptors and phorbol esters (*Cavadas et al., 2017*; *Bleibaum et al., 2019*). iRhom2 does not control the trafficking to the cell surface; rather, phosphorylated iRhom2 controls the rapid stimulation of TACE activity (*Lee, Nam & Choi, 2016*). Another study showed that GPCR in histamine agonists triggers the TACE-dependent release of EGFR ligands like TGF α and amphiregulin in an iRhom2 phosphorylation-dependent manner (*Grieve et al., 2017*).

Thus, it can be anticipated that during DCIS, grade 2 and grade 3, the isoform switches from canonical to cub isoform where the cells need to proliferate cancer growth *via* EGFR signalling pathway and the pathway is upregulated by the indirect activation of TACE by GPCR agonists. In contrast, this phenomenon decreases when cells undergo metastases, where the primary task is invasion, and the isoform switches back to canonical, as observed during CTCs.

## CONCLUSIONS

In our study, pathways leading to TACE-dependent EGFR signalling pathways were more observant; specifically, MAPK signalling pathways, GPCR signalling pathways, and toll-like receptor pathways in DCIS, grade 2 and grade 3. However, no direct relationship was found with EGFR or ERAD for iRhom2 or its interacting partners. Nevertheless, it is noteworthy that during CTCs, the cub isoform switches back to the canonical isoform. Furthermore, in addition to the processes mentioned above, the proteasomal degradation pathway and cytoplasmic ribosomal protein pathways were significantly enriched. Therefore, it could be inferred that both the isoforms have separate physiological roles during tumorigenesis.

### Funding
This work was supported by the HEC NRPU (project id 9988). The funders had no role in study design, data collection and analysis, decision to publish, or preparation of the manuscript.

### Grant Disclosures
The following grant information was disclosed by the authors:
HEC NRPU: 9988.

### Competing Interests
The authors declare that they have no competing interests.

### Author Contributions
- Mehar Masood performed the experiments, analyzed the data, prepared figures and/or tables, authored or reviewed drafts of the article, and approved the final draft.

- Madahiah Bint E Masood analyzed the data, authored or reviewed drafts of the article, and approved the final draft.
- Noor Us Subah performed the experiments, analyzed the data, authored or reviewed drafts of the article, and approved the final draft.
- Maria Shabbir conceived and designed the experiments, authored or reviewed drafts of the article, and approved the final draft.
- Rehan Zafar Paracha analyzed the data, authored or reviewed drafts of the article, and approved the final draft.
- Mehak Rafiq conceived and designed the experiments, prepared figures and/or tables, authored or reviewed drafts of the article, and approved the final draft.

## Data Availability

The code is available from Vitting-Seerup K, Sandelin A. 2017. The Landscape of Isoform Switches in Human Cancers Mol Cancer Res 15 (9): 1206–1220. https://doi.org/10.1158/1541-7786.MCR-16-0459.

## Supplemental Information

Supplemental information for this article can be found online at http://dx.doi.org/10.7717/peerj.14124#supplemental-information.

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
