# Peer review of "Investigating isoform switching in RHBDF2 and its role in neoplastic growth in breast cancer"

_PeerJ, doi:10.7717/peerj.14124_

## Round 0.1 · original submission · Major Revisions

After careful consideration and weighing in the reviewers' decision, I have to ask the authors for a complete major revision of the manuscript and a complete rewrite.

The manuscript aims to show whether an isoform switch happens during breast tumour progression aiming at one particular gene (RHBDF2).

I would like to emphasise checking Reviewer 3's valid comments and Reviewer 4's suggestion to validate the finding with an additional modality.

I have a few comments regarding the manuscript myself:

The authors described why they have chosen this gene, but sadly only in the discussion. Please move the role of RHBDF2 into the introduction.

Please describe which type/subtype of breast cancer were used in the analysis (LuminalA/B, triple-negative, HER2 enriched etc.).

Figure 5 with the actual sources of the experiments can be useful to understand what comparisons the authors made.

I would like to see the citation and version or download date of every used tool.

Please define every acronym in the manuscript.

Reviewer 1 ·

Basic reporting

No comments

Experimental design

No Comments

Validity of the findings

No Comments

Additional comments

Overall the paper reflects a strong hypothesis and the study is well designed.

·

Basic reporting

I think this manuscript is readable, well-written, and presents a logical order. I just have some comments:

- The introduction lacks references for many clauses. Some examples were highlighted in the attached file.

- Please, use standard formatting rules for gene names. They should be shown in italic.

- You should include the version for R (and other) used software, please. In line 110, I think it should say "using the isoformSwitchAnalyzeR package".

Experimental design

- Datasets are stored at Genbank servers. How did you download it from EMBL-EBI? Please, explain it.

Validity of the findings

- In Figure 1, It would be interesting to mention which groups are represented for each color in each comparison.

- Regarding data reliability, I suggest sending your pre-processed data if available.

Reviewer 3 ·

Excellent Review

This review has been rated excellent by staff (in the top 15% of reviews)
EDITOR COMMENT
The reviewer pointed out flaws in the manuscript. The review was thorough and respectful toward the authors, and showing particular technical skills. The reviewer's willingness of review a revised manuscript made the hard decision easier. I am grateful for their help and comments.

Basic reporting

The current study focuses on iRhom2 isoform switch and its role in breast cancer initiation and metastasis. The research motivation comes from iRhom2’s paradoxical interactions with other signaling pathways, especially EGFR pathway, as reported by previous literatures. After studying expression patterns of iRhom2’s canonical isoform and cub isoform across different breast cancer subtypes and stages, the authors propose the Cubs pathway to explain the mechanism of action of iRhom2’s isoforms in tumorigenesis.

In general, the manuscript needs to be improved in English written. The abstract and introduction may, particularly, need rephrasing to make the statements they make unambiguous. Attention should be paid when pronouns and clauses are used to refer between entities. Some minor typos and format issues also need to be corrected, including but not limit to, Line 54 “member’s” to “members”, Line 271 citation author name needs to be capitalized, Line 310 space is needed after the citation. Full spellings should also be provided before acronyms are used, especially for short acronyms that may lead to polysemy (e.g., Line 276 AS analysis tool, Line 290 KO for knockout). For some cases, the full name comes after the acronym has been used, like Line 317 TNF-α converting enzyme for TACE, where TACE has appeared in the previous text.

The introduction tries to introduce the topic of the study by fast narrowing down the scope from breast cancer to Rhomboids and alternative splicing, but it does provide enough background details or smooth logic transitions to make the research question clear. Rather, a straightforward introduction that goes directly into the problem of the heterogeneous roles that iRhom2 isoforms play may be more helpful.

The Discussion section, on the other hand, contains two many details. The first few paragraphs serve more like a literature review, from breast cancer, to alterative splicing, to the function of iRhom2 and its regulation on EGFR signaling pathway, rather than a typical result “discussion” that should be included in this section. The cited literatures are comprehensive, yet not all closely related to the discoveries in this study or help derive the conclusion in this study. The authors may consider moving some parts directly related to iRhom2 to Introduction instead.

Experimental design

The question to be answered by this study is not clearly defined, or maybe well-defined but poorly conveyed to general readers due to the English language being used. Yes, the proposed work section states the topic of the study, but for a data-analysis based article (or any experiment-based article), the exact question to be answered should be clearly defined and emphasized from the beginning (e.g. using a computational pipeline to compare the difference in the expression level of two iRhom2 isoforms across different breast cancer subtypes and stages).

The data processing pipeline introduced in Methods is solid for processing RNA-seq data, but a lack of innovation for typical computational biology research (though this is not required). It is recommended to include a detailed introduction in Methods of what hypothesis tests were used to compute p-values reported in the study (e.g. p-value mentioned in Line 142, Line 193, and p-values reported in Table 3 and Table 4), what type of correlations were used (Pearson? do not just refer to it as “correlation analysis” as there are at least several types of correlations that are commonly used in computational biology), and what standards were used to determine if an observation is significant.

Validity of the findings

The study used public data and public packages only, thus no raw data or code is provided. However, it is recommended to provide intermediate results to make sure the experiment is reproducible. One important intermediate dataset that is missing is the gene set identified with correlation analysis for each GSE dataset which was then used for GSEA. Please provide these gene sets as supplementary materials.

Though samples are not directly compared across data sources, it would still be wise to do a batch effect analysis for the raw datasets used in this study. For example, the expression level of the same isoform in normal samples is very different across datasets according to Figure 1. Is that because of the batch effect or something else?

Many of the analyses conducted in this study will be affected by sample size, but the sample size information is not mentioned or provided for the raw datasets being used. For example, Line 203 mentions the IF value was just averaged over the two primary tumor datasets and two IDC datasets. This is inappropriate without considering the sample size of the two datasets (and without considering the potential batch effect).

The major discovery or proposal of this study is the Cubs pathway, which is based on the GSEA and leading-edge analysis results, which are further based on the gene sets identified using the correlation analysis (which is itself not clearly defined in this study). However, it is known that GSEA and leading-edge analysis are not robust statistical analyses, as they are very sensitive to the candidate gene set size and pre-defined gene group size. It is recommended that GSEA should only be used to provide general insights about what potential biological process a candidate gene set may be representative, and it should not serve as the only foundation for deriving a new signaling pathway. If proposing the Cubs pathway remains to be the main objective of the study, then it is strongly recommended that the authors could validate their pathway map with wet-lab experimental results or at least develop more robust computational approaches for expression pattern analysis. In that case, the introduction of the Cubs pathway should be moved from Discussion to Results as it will be the directly validated outcome of the study, rather than simply a conjecture.

Other minor changes required include: color scheme legend is needed for Figure 1.

Reviewer 4 ·

Basic reporting

In this manuscript, the authors offered a new perspective to study on proteins that have opposing roles in tumorigenesis and had taken good advantage of bioinformatics tools and public dataset to reach to their goals. However, with the current analysis, the authors merely stated a phenomenon that RHBDF2 will switch to different isoforms during different stages through breast cancer development and provided no solid data to on how this switch would affect breast cancer progression. This reviewer could not tell whether the isoform switching is caused by breast cancer deterioration or leads to it, not to mention if it could serve as a target for cancer treatment. The authors need to do some molecular and biochemistry assays to prove the existence of a cause-and-effect relationship.

Experimental design

N/A

Validity of the findings

N/A

---

## Round 0.2 · Minor Revisions

Please check the reviewer's suggestions and address the minor changes.

·

Basic reporting

Dear authors,

Thank you for addressing my previous concerns. However, I noticed that you attached the original version of the manuscript instead of the updated one. I was able to check the edits only in the change tracking file. However, this file lacks figures. Therefore, I still do not have enough material to support this manuscript. Could you attach the correct file to the platform?

Experimental design

no comment

Validity of the findings

no comment

Reviewer 3 ·

Basic reporting

The current revision has addressed many issues mentioned by reviewers previously. Some, however, remain unattended and some further improvements may be required as listed below.

The newly uploaded PDF manuscript is the same as the last version except for removing the Acknowledgement section. None of the changes made on the manuscript track changes word file is reflected in the PDF version. Please make sure the right manuscript is uploaded.

The claim of the “stressing over the assumption of a ‘one gene to one protein to one functional pathway’” in line 110 (track changes doc) may be too strong. It could be the case that most studies assume one gene leads to one protein, but it is rare that current cancer studies assume only one protein matters in a whole signaling pathway. The authors may consider rephrasing this claim.

Line 126 “iRhom2 an interesting gene to study”, iRhom2 is the protein name, not the gene name.

There is no updated PDF version uploaded so it is hard to check all the text format issues. But at least based on the word version, missing and abundant spaces can be found over the manuscript, as well as font inconsistencies.

Experimental design

Thank the authors for the revision effort, the overall experimental design has been improved significantly.

Validity of the findings

The relative expression levels of canonical and cub isoforms in normal samples are very different across datasets in Figure 1 (some with canonical higher like GSE110114, some with cub higher like GSE130660, and some with relatively the same fraction of expression like GSE69240). Since they are all supposed to be normal tissue samples, any explanation why there is such a variance? Otherwise, if the baseline (normal samples) could vary so much across data sources, it is hard to justify that the isoform switching patterns the author observed are truly resulted from isoform switching over different stages of tumor development or just from batch effects existing among the datasets. This is more concerning, knowing that some of the datasets like GSE130660 are actually quite small.

May need more explanations on how Figure 2 is generated. The other bars seem to be consistent with Figure 1, but the Cub bars of IDC, G2, G3, and CTCs are not aligned with Figure 1. For example, according to Figure 1 F, Cub increases at least 0.2 from normal to grade 2 (red bar > 0.5 and green bar = 0.3) but in Figure 2 the increment is only 0.06. If this is actually correct, then the “Differential Isoform Usage” section may need some rephrasing to clarify the approach used for generating Figure 2.

Gene name missing (NKFB1?) on line 394 in “whereas is the only interesting gene found in 2…”.

Line 661 right parenthesis missing.
Line 663 leading left parenthesis seems to be redundant.

It is highly recommended that the authors split the last long paragraph in the Discussion into several paragraphs to reduce the scope of topics each covers. In addition, it may not be necessary providing detailed descriptions of every pathways that involved with iRhom2 isoform switching. Space could be largely saved by focusing only on the mechanism of the Cub pathway the authors want to propose.

Reviewer 4 ·

Basic reporting

The authors' reply to this reviewer's concerns is reasonable and acceptable. No further revision is required.

Experimental design

N/A

Validity of the findings

N/A

Additional comments

N/A

---

## Round 0.3 · accepted · Accept

The authors addressed the reviewer's comments.

·

Basic reporting

no comment

Experimental design

no comment

Validity of the findings

no comment

Reviewer 3 ·

Basic reporting

Thank the authors for their efforts in revising their manuscript in addressing the reviewer's concerns. No further revision is required.

Experimental design

NA

Validity of the findings

NA